# The Key Role of Nutritional Elements on Sport Rehabilitation and the Effects of Nutrients Intake

**DOI:** 10.3390/sports10060084

**Published:** 2022-05-26

**Authors:** Sousana K. Papadopoulou, Maria Mantzorou, Foivi Kondyli-Sarika, Ioanna Alexandropoulou, Jannis Papathanasiou, Gavriela Voulgaridou, Pantelis T. Nikolaidis

**Affiliations:** 1Department Nutritional Sciences and Dietetics, International Hellenic University, 57400 Thessaloniki, Greece; sousana@the.ihu.gr (S.K.P.); fiviks@gmail.com (F.K.-S.); ialexand@med.duth.gr (I.A.); gabivoulg@gmail.com (G.V.); 2Department of Food Science and Nutrition, University of the Aegean, 81400 Lemnos, Greece; mantzorou.m@aegean.gr; 3Department of Medical Imaging, Allergology& Physiotherapy, Faculty of Dental Medicine, Medical University of Plovdiv, 4002 Plovdiv, Bulgaria; giannipap@yahoo.co.uk; 4Department of Kinesitherapy, Faculty of Public Health, Medical University of Sofia, 1431 Sofia, Bulgaria; 5School of Health and Caring Sciences, University of West Attica, 12243 Athens, Greece

**Keywords:** athletes, carbohydrates, energy, lipids, minerals, protein, rehabilitation

## Abstract

Adequate nutrition is of utmost importance for athletes, especially during rehabilitation after injury in order to achieve fast healing and return to sports. The aim of this narrative review is to define the proper nutritional elements for athletes to meet their needs and facilitate their fast return to sports after surgery or injury, as well as determine the effects of specific nutrients intake. Studies on antioxidants, which are substances that protect against free radicals, for the injured athlete are few and unclear, yet poly-phenols and especially flavonoids might improve healing and inflammation following an injury. Benefits of vitamin C or E on muscle damage are disputable in relevant studies, while optimal levels of vitamin D and calcium contribute to bone healing. Minerals are also essential for athletes. Other supplements suggested for muscle damage treatment and protein synthesis include leucine, creatine, and hydroxymethylbutyrate. Diets that include high-quality products, rich in micronutrients (like vitamins, minerals, etc.) bio-active compounds and other nutritional elements (like creatine) are suggested, while an individualized nutrition program prescribed by a trained dietitian is important. Further studies are needed to clarify the underlying mechanisms of these nutritional elements, especially regarding injury treatment.

## 1. Introduction

Sports injuries have a significant economic [1] and psychological [2,3] impact and could affect physical capabilities [3]. Some injuries have no impact on future quality of athletes’ life, but others can negatively affect it even if only for a short period of time [3]. The incidence of injury in athletes was 81.1 injuries per 1000 participants in International Championships [4]. Thirteen top-level championships are included in that International Championship and particularly seven outdoor championships (e.g., Olympic Games during summer of 2008 and 2012 and winter of 2010), four indoor championships, one world youth championship (aged 16–17 years), and one world junior championship (16–19 years). The sports categories included in these championships were distance sports (short, middle, or long), “jumps”, “race walks”, “throws”, and “combined events (e.g., pentathlon). Further, the World Indoor Championships had a marathon category. Indoor championships include fewer categories [4].The incidence of injury in athletes varies according to (a) the injury time (72.2% during competition and 21.8% during training), (b) the type of championships (higher in outdoor than in indoor championships or in youth/junior championships), (c) the discipline categories (highest in combined events followed by Marathon and then long and middle distances) [4], and (d) the gender (higher in males than females) [5]. The most common injuries are muscle injuries [6] and bone injuries like stress fractures and tendinopathy mainly in high-jerk sports [7].

Prolonged injury leads to sedentary periods that cause a decrease in muscle mass, muscle strength, and function [8]. In addition, muscle atrophy and increased abdominal fat deposition may delay further the return to competition [9]. There is a lack of evidence in the literature about the role of micronutrients and supplements during injury though there is encouraging indirect documentation in muscle recovery and sarcopenia [9]. Sarcopenia is mainly a disease that occurs in the elderly; however, its development may be related to conditions that do not only affect the elderly, such as malnutrition, disuse, and cachexia [10].

Rehabilitation refers to the techniques and services for individuals with physical disabilities (e.g., after injuries, stroke etc.) trying to make them reach and keep a functional level in their everyday life as normally as possible [11]. Sports rehabilitation is the therapeutic approach that targets athletes’ recovery, treating their pain, and helping them return safely to sport [12]. On the other hand, sports recovery includes strategies applied immediately post-exercise aiming to recover glycogen content and from muscle damage in order to maximize their performance while minimizing the risk of injuries [13,14,15].

The use of sports rehabilitation strategies is essential for a fast rehabilitation process in athletes’ health and their return to training and competition. In the early stages of rehabilitation, nutritional intervention is crucial to ensure adequate intake of energy and nutrients, prerequisites for wound healing and management of inflammation and oxidative stress caused by the injury [15].

Adequate and balanced nutrition is essential for rehabilitation progress [15]. The role of energy and macronutrients requirements in rehabilitation has already been extensively discussed in a previous study [15]. Thus, proper nutrition for injury recovery of athletes requires high consumption of energy 25–30 kcal/kg of body weight as well as adequate intake of carbohydrates and especially proteins (type, frequency and amount) to support athletes anabolism [15]. On the other hand, the role of micronutrients and other nutritional elements (except macronutrients) in athletes’ rehabilitation is an underdeveloped area with enormous potential regarding the time loss from training and competition due to injuries. This review aims to define the proper nutritional elements tailored by athletes’ needs in order to facilitate their fast return to sports after surgery or injury.

## 2. Supplements of Micronutrients and Other Bioactive Compounds

### 2.1. Carotenoids and Polyphenols

The role of vitamin and other micronutrients on sport rehabilitation is presented in Table 1. There are approximately 600 carotenoids found in fruits and vegetables, with *β*-carotene being one of the most common [16]. Some carotenoids are precursors of vitamin A and others act as antioxidants [17]. Recent reviews did not demonstrate the benefits of taking isolated *β*-carotene supplements. Instead, it is widely suggested that *β*-carotene should be consumed through carotenoid-rich foods, as part of a diet that includes consisting of the appropriate macro-, micronutrients and other nutritional elements [18]. Findings from various studies strongly support the anti-inflammatory and antioxidant potential of carotenoids’ intake for their protective effect on the development of several health problems, which are mediated by oxidative stress, caused by an increase of reactive oxygen species (ROS) [19] (Table 1). The antioxidant and anti-inflammatory actions of phenols, carotenoids, and tocopherol, contained in olive oil, have also been associated with protective effects against common chronic diseases. Their interaction with the inflammatory progress has been shown to prevent cellular damage [19]. Epidemiological studies have shown that carotenoids and polyphenols are inversely associated with sarcopenic symptomatology; thus, they are likely to preserve skeletal muscles and function [20]. In a recent 12-week intervention, an energy restricted diet, to achieve a weight loss of half-to one kg/week, was prescribed to obese or overweight healthy older adults followed macronutrient distribution of 30% protein, 30% carbohydrate, and 40% fat [21]. They were divided into two groups to follow a diet of vegetables and fruits with either high carotenoid content and extra virgin olive or low carotenoid content and polyunsaturated fatty acid based oil [20]. A positive change in appendicular skeletal muscle (ASM) indicated the synergistic effect of nutrients and the potential influence of high carotenoid and polyphenol consumption on body composition and skeletal muscle function [21].

Polyphenols such as resveratrol and flavonoids such as quercetin and catechins have also caught scientists’ attention the last decade for their protection of muscles against exhaustive exercise and oxidative stress [39]. However, for each bioactive compound there is no cause–effect potential benefit associated with muscle mass, strength, and physical performance [20]. Flavonoids, though, are likely to accelerate healing, in cases of injury and inflammation, through their antioxidant actions. As it was observed by Skarpańska-Stejnborn et al., flavonoids included in daily diet may be beneficial in cases of severe injury—an assertion which is based on their role in regulating the immune factors of the inflammatory procedure and the sequestration of iron [21]. Moreover, polyphenols like oleuropein may reduce the development of chronic joint inflammation, swelling, pain, and other clinical indications, in relation to rehabilitation or prevention of osteoarthritis [40]. The benefits of phenolic compounds combined to physical activity on joint disease, consist a positive effect on homeostasis of cartilage tissue after injury [30].

### 2.2. Vitamin D

Worldwide, it is estimated that one billion people have vitamin D insufficiency or deficiency [41]. Optimal levels of vitamin D and calcium have been recognized for their contribution to bone healing, which is important for the rehabilitation of athletes [42]. Vitamin D seem to have a positive impact on muscle growth and cell differentiation, and further it could also increase sarcoplasmic calcium uptake leading to higher muscle contractility [43].

In vitro studies have shown that vitamin D receptors (VDR) are expressed in the muscle stem cells depicting muscle regeneration after injury [44]. A recent review revealed that vitamin D supplementation from any source had no effect on muscle mass in elders, even though it benefitted cultured muscle fibers cells in vitro [45].

Vitamin D supplementation has different outcomes according to muscle groups and functions [46]. A meta-analysis of randomized controlled trials in elders depicted that vitamin D had a beneficial effect on muscle strength but no impact on muscle mass and power [47]. Similarly, Zhang et al. recently found in athletes that vitamin D had a beneficial effect on lower-limb muscle strength but not upper-limb muscle strength or overall muscle strength and muscle explosive power [46]. In a meta-analysis of randomized controlled trials, vitamin D supplementation in healthy adults was significantly associated with increased upper and lower limb strength [29] (Table 1). The impact of vitamin D supplementation on skeletal muscle mass and function is controversial and further research is needed.

### 2.3. Vitamins C, E and A

Vitamins A, E and C have frequently been called “antioxidants” since their deficiency tends to lead to oxidative stress [48]. There is also growing support for their benefits, particularly during rehabilitation. Foods containing vitamin C and flavonoids might be beneficial during the rehabilitation process when the body regenerates tissue [25]. Randomized controlled trials did not support vitamin C supplementation in the general population [49], yet it may be of use in surgery patients, whose vitamin C requirements are increased [26].

Vitamin E supplementation in rats following spinal cord injury improved the hind limb locomotor function, and reduced spinal cord histopathological and morphological damage, while decreasing inflammation [28]. Belisle et al. found in elderly that vitamin E supplementation may enhance cytokine IL-1β, IL-6, TNF-α, and IFN-γ production, but this depends on immune defense system status of each individual [50] (Table 1). Supplementation with vitamin C above sufficiency, however, is not suggested and yet there is no strong evidence, in general, for the necessity of this specific micronutrient supplementation after injury [51]. During rehabilitation, while the antioxidant intake is necessary for best results, overconsumption is not recommended when nutrient indicators are at normal levels [29]. Supplementation with mega-doses of vitamins A and E may hinder the improvements in areal bone mineral density after strength training [52].

Vitamin A is an essential fat-soluble vitamin that has been used topically in dermatology for several years to control conditions such as photo-damage, psoriasis, and after procedures such as dermabrasion to facilitate healing [53] (Table 1). Anti-inflammatory effect in open wounds has been found to stimulate epithelial development, fibroblasts, and ground material [53]. The literature confirms the beneficial impact of supplementary vitamin A in acute wounds and in the healing of injuries caused in bones, burns, intestines, and radiation [22]. An earlier study by Hunt et al. [23] on the effect of vitamin A on reversing the inhibitory effect of cortisone on healing of open wounds in animals and humans showed that topical vitamin A can revert the cortisone-related healing inhibition in open wounds without affecting other open wounds that are not treated with vitamin A in the same patient or animal, and that the antagonism between vitamin A and cortisone often occurs in humans and rabbits and rats. Vitamin A tends to serve as a hormone that alters the development of epithelial cells, melanocytes, fibroblasts, and endothelial cells via a family of receptors for retinoic acid [24]. During the rehabilitation phase, it seems that some vitamins may be needed for the appropriate nutritional support.

## 3. Supplements of Nutritional Elements

### 3.1. Creatine

The results of creatine on muscle strength and muscle mass gain have been examined regarding muscle atrophy, during immobilization or rehabilitation. Evidence for the impact of creatine on muscle loss during immobility is not very clear [31]. As shown in young men, 7-day creatine supplementation at 20 g/day, which is generally considered as a “high” or initial “loading” dose, could decrease loss of muscle mass and strength after immobilization of the upper arm [32]. Arm and leg muscles, however, seem to react differently; thus, creatine supplementation after surgery decrease atrophy in immobilized arm [32] while it did not lessen muscle loss during lower-limb immobility [33]. On the other hand, other studies reported increased muscle growth and strength after immobility and during rehabilitation, via creatine supplementation, indicating that it may have a protective effect on muscle damage in cases of muscle inactivity during injury rehabilitation [54] (Table 1). As Hespel et al. [33] reported, muscle hypertrophy was stimulated in individuals with 2-week leg immobilization and a 10-week rehabilitative strength training program, by using oral creatine monohydrate from 20 g down to 5 g daily. Specifically, quadriceps muscle cross sectional area was increased by 10% accounting for hypertrophy of both type I and type II muscle fibers and peak strength by 25% (Wmax knee extension) and these changes were linked to myogenic protein expression and regulating factor 4 (MRF4). In a similar study, healthy individuals in a 2-week leg immobilization participated in a 10-week long heavy resistance training program with oral creatine supplementation. Results showed that the intake of creatine offsets the decline in muscle GLUT4 protein during immobilization and increased GLUT4 protein content during rehabilitation, increasing insulin sensitivity, exerting a beneficial effect on glucose homeostasis throughout the body and thus increasing glucose uptake into muscle [55].

Moreover, creatine supplementation at 20 g/day for 7 days has been shown to strengthen exercise capacity in individuals with complete cervical-level spinal cord injury [56]. Conversely, in patients with anterior cruciate ligament reconstruction, the intake of creatine at 20 g/day for 7 days and 5 g/day thereafter did not show benefits, during a period of 12-week rehabilitation [57]. The safety of creatine supplementation has also been assessed in several studies. According to the official position of the International Society of Sports Nutrition, the ergogenic benefits of creatine monohydrate, which is the most clinically effective and extensively studied type of creatine, remain consistent in scientific evidence. In addition, other potential health benefits from creatine intake have been reported. Thus, creatine monohydrate supplementation is considered to be a safe choice and the most effective ergogenic nutritional supplement for athletes. It is also safe for patients or healthy people of all ages, as long as precautions and supervision are provided [54,55].

### 3.2. Gelatin and Vitamin C/Collagen

Early research data on the prevention and treatment of injuries to tendons and ligaments highlight the role of nutritional support from collagenous proteins and micronutrients such as vitamin C, which have shown potential benefits [58] (Table 1). The recommended dosage is 5–15 g gelatin with 50 mg vitamin C. Collagen hydrolysate dosage is 10 g/day. Increased collagen production, thickened cartilage, and decreased joint pain are some of the benefits that have been observed, while the use of gelatin and collagen supplements appears to be of low risk [27,59]. Few data are available, yet increased collagen production and decreased pain may be possible benefits. Functional benefits, rehabilitation from injury, and effects in elite athletes are not known yet.

### 3.3. Minerals

Minerals, such as manganese (Mn), copper (Cu), zinc (Zn), iron (Fe), and selenium (Se), act as antioxidants and, specifically as co-factors of important antioxidant enzymes, inactivate ROS and repair oxidative damages [34,35] (Table 1). Adequate intake of minerals like magnesium (Mg), iron (Fe), and selenium (Se) is essential for athletes as they are involved in muscle contraction, oxygen transport, antioxidant capacity and many other functions necessary for optimized health as well as peak athletic performance [60,61,62,63]. Supplementation is deemed necessary in the presence of deficiency [59,62,64,65]. Iron’s role is important for oxygen transport and perfusion of the tissues, energy metabolism, antioxidant processes and collagen synthesis [66,67]. Therefore, its deficiency, which is common among athletes, could lead to tissue ischemia, impaired collagen production, and decreased wound strength in the healing phase; thus, it can become a severe health risk factor [67,68]. It must be noted, though, that iron supplementation should be strictly monitored, as it could produce several side-effects; thus, a diet rich in iron is recommended. More specifically, the consumption of animal protein like meat and fish, dairy, fruits, and vegetables (particularly of the green leafy kind) could balance the dietary needs for the nutrients that contribute in a healthy body condition [68]. The current instructions regarding this recommended dietary intake of iron is 8 mg per day for males and 18 mg per day for pre-menopausal adult women [69]. During inflammation, zinc has promotive role for the response of the immune system and the cell proliferation phase, due to its beneficial role during DNA replication in cells in proliferation phase [67]. Selenium has also been found to participate in wound healing as a reducing factor of oxidative stress [70]. More specifically, glutathione peroxidase (GPX) is the main selenoprotein in the human body, having the role of inhibiting the overproduction of free radicals at the site of inflammation, along with other selenoproteins [70]. Recommended daily intake of selenium has been determined as 55 μg per day in adult males and 70 μg per day in adult females [70]. As a result, all the above minerals seem to be of crucial importance in an athlete’s diet.

### 3.4. Omega-3 Fatty Acids

Unsaturated fatty acids have anti-inflammatory and immune-modulatory effects. A prolonged inflammatory reaction, often seen after a serious injury or surgery, can be detrimental to rehabilitation. It has been suggested that n-3 PUFA lead to a decrease of muscle inflammation through the reduction of the level of pro-inflammatory cytokines, such as TNF-α and IL-6, reduced production of arachidonic acid (AA) and reactive oxygen species (ROS), resulting in a reduction in the inflammatory response [71]. Fish oil-derived omega-3 fatty acids have been found to regulate muscle protein synthesis [72]. More specifically, a long-term intake of omega-3 fatty acids (4 g/day) benefits anabolic sensitivity to amino acids [73]. Although it is stated that, omega-3 supplementation probably ameliorates muscle loss during immobilization, it has not been proven effective for muscle gain. This is why further discussion is required regarding the appropriate dose for injury rehabilitation and prevention.

### 3.5. Anti-Inflammatory Supplements

Curcumin, a constituent of the spice turmeric, is a highly pleiotropic molecule with anti-inflammatory and anti-oxidant properties that have shown a variety of beneficial effects on human health [36]. A systematic review in patients with arthritis showed that curcumin reduced symptoms of inflammation and pain, with similar outcomes to ibuprofen and sodium diclofenac [74]. Decreases in inflammatory cytokines and other indirect markers of muscle damage with anti-inflammatory supplements such as curcumin and tart cherry juice have been reported [37] (Table 1). Heaton et al. [75] state that there is limited evidence for the role of curcumin at a dosage of 0.4–5 g/day, and bromelain, a proteolytic enzyme found in both the stem and fruit of pineapple, at a dose of 900–1000 mg/day in decreasing inflammation, which is important for athletic rehabilitation. Curcumin has shown benefits regarding postoperative pain and fatigue [38]. More research is needed before these compounds can be recommended to athletes.

In general, a multi-disciplinary team, consisting of a doctor and a nurse, a physiotherapist, and a dietitian are necessary for optimal rehabilitation. The role of the dietitian is crucial, to plan an individualized nutrition program, that takes into consideration the athlete’s needs and preferences, and according to the current guidelines and recommendations, regarding nutrition during rehabilitation. Micronutrients, such as carotenoids, polyphenols, flavonoids, vitamin E or C are greatly supported for their properties against oxidative stress and they might have anti-sarcopenic properties. Although findings on the effects of antioxidants for the injured athlete are few and unclear, it has been shown that polyphenols and especially flavonoids might improve healing and inflammation following an injury. Additionally, optimal levels of vitamin D and calcium contribute to bone healing, thus they are also beneficial during the rehabilitation process. Furthermore, both the antioxidant and the anti-inflammatory properties of ingredients such as omega-3 fatty acids and curcumin could be beneficial during athletic rehabilitation. Other supplements suggested for muscle damage treatment and protein synthesis include leucine, creatine and HMB. Mineral with antioxidant function, such as manganese (Mn), copper (Cu), zinc (Zn), iron (Fe), and selenium (Se), are also essential for athletes. Athletes’ requirements and supplementary doses should be considered carefully, preferably through a diet that includes high-quality products, rich in nutrients.

## 4. Nutritional Elements and Specific Injuries

### 4.1. Rehabilitation of Muscle Injuries

The anti-inflammatory effects of the Omega-3 polyunsaturated fatty acids (n-3 PUFA) have been observed to attenuate during various rehabilitation interventions of muscle injuries. The level of supplementation, in this case, should remain much higher than in the case of a typical diet [76].

### 4.2. Rehabilitation of Bone Injuries

Rehabilitation of bone injuries after an athletic injury may benefit by sufficient intake of nutrients like calcium, and vitamins like D and A, along with other macronutrients (proteins) [42]. Furthermore, there is strong evidence that the presence of adequate levels of manganese, copper, boron, iron, zinc, silicon, vitamin A, vitamin K, vitamin C, and the B vitamins are of great importance for the support of the bone tissue [77].

### 4.3. Rehabilitation of Tendons and Ligaments

Close et al. conducted an extended study regarding the micronutrients that promote the health and rehabilitation of tendons and ligaments. The main micronutrients in that area are vitamin C, copper, glycine, gelatin/hydrolyzed collagen, turmeric/curcumin, taurine, arginine, bromelain, or boswellic acid [38,68].

## 5. Conclusions

During various exercise-based rehabilitation interventions, high-quality proteins, fat, vitamins, antioxidants, minerals, and other supplements can play a major role in supporting athletes’ anabolism. Although there are several studies on the ergogenic effect of nutrients and supplements before and during training and competition, there is a research gap on the effectiveness of these nutrients in the rehabilitation of athletes after injury or surgery, in maintaining muscle mass and in reducing rehabilitation time. For this reason, it is not possible to make definitive recommendations on the use of the nutrients and supplementations. Future research is warranted to clarify the underlying mechanisms of nutrients, especially regarding injury treatment, as their efficacy has not yet been assessed satisfactorily. Monitored evaluation should follow, in order to assess nutrient indicators and to avoid levels above sufficiency. Diets that include high-quality nutrients, rich in macro- and micro- and bioactive compounds are suggested. Biomedical indices and vitamin and mineral levels should be assessed and monitored, to avoid unnecessary supplementation.

## Figures and Tables

**Table 1 sports-10-00084-t001:** The role of vitamins and other micronutrients on sport rehabilitation.

Nutrient	Function on Rehabilitation	References
**Vitamin A**	Positive impact on acute wounds Positive impact on healing of fractures Hormone action in altering the activity of epithelial cells, melanocytes, fibroblasts, and endothelial cells via retinoic acid receptors Counteraction of corticosteroid impact on wound healing	[22,23,24]
**Vitamin C**	Regulation of cytokines and oxidative stress Collagen formation Adequate levels result in improved muscle strength after surgery	[25,26]
**Gelatin and vitamin C/collagen**	Increased collagen production Thickened cartilage Decreased joint pain	[27]
**Vitamin E**	Decreased inflammation Improvement of limb function Reduction of spinal cord histopathological and morphological damage	[28]
**Vitamin D**	Bone formation and healing Increased upper and lower limb strength Increases in type II muscle fibers and muscle strength Improvements in atrophy	[25,29]
** *Carotenoids, Polyphenols and Flavonoids* **	Anti-inflammatory action Homeostasis of cartilage tissue after injury Decreased sarcopenicsymptatology Increased appendicular skeletal muscle	[20,30]
** *Creatine* **	Decreased muscle mass loss & decreased strength loss after immobilization of upper arm Increased muscle growth and strength after immobility and during rehabilitation Protective effect on muscle oxidative damage	[31,32,33]
**Minerals** **Manganese (Mn), Copper (Cu), Zinc (Zn), Iron (Fe), Selenium (Se)**	Antioxidant function Repair of oxidative damage	[34,35]
**Anti-inflammatory supplements**	Anti-inflammatory and anti-oxidant functions regarding muscle damage and post-operative pain and fatigue Curcumin improves arthritis outcomes	[36,37,38]

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
