# Peer review of "The Key Role of Nutritional Elements on Sport Rehabilitation and the Effects of Nutrients Intake"

_sports, 2022, doi:10.3390/sports10060084_

Round 1
Reviewer 1 Report
The title suggests that the role of micronutrients will be covered in the manuscript but there is also communication about leucine (an amino acid) and creatine, for example. Please reconsider the title.
The manuscript would benefit from a definition/description of micronutrients to be clear for the reader what nutritional elements are considered micronutrients. Is there consensus that creatine and leucine are considered micronutrients?
I feel that there is a lack of focus in the review and most of the information seems to be deal with effects of nutrients intake.
There seems to be a tendency not to use primary sources, e.g. ref 6 (L51). Please use original sources or clarify that review articles were referenced.
Is recovery the same as rehabilitation. Recovery post-exercise is short lasting. Please be consistent in the manuscript.
L23. “rich in nutrients”. Please clarify what kind of nutrients. Do you mean rich in particular micronutrients?
L25. “mechanisms of nutrient”. Do you mean micronutrients?
L39. Reference 7 has no information on stress fractures and tendinopathy. Please check.
L42. Ref 9. I suggest to replace with an English reference.
L44. Is there consensus that changes in muscle by prolonged injury are considered a sarcopenic event. Sarcopenia seems to be reserved mostly for aging-induced changes in muscle mass and function. Please reconsider.
L53. Ref 11 is on macronutrients but also cover leucine for example. Is there replication?
L59. Considering the aim of the review, why is there no advice on required micronutrient intake.
L64. No information in vitamins is presented in this section.
L69. What are “high-quality products”. Please clarify and provide an example of such a product.
L73. “overproduction” to get oxidative stress but we normally always experience some level of oxidative stress. In addition, ref 15 does not provide experimental observations on overproduction of ROS. Please clarify/revise.
Ls 77-89 is on aging and weight loss. The authors should focus their writing on rehabilitation and micronutrients.
L92. Recently?, is 2013 recently?
L95. “Flavonoids should be included in daily diet”. What is the recommendation considering it is suggest to have higher intake with severe injury. Please provide original references to support this assertion.
L106. Ref 20 is not an original source to support the statement. Please provide an original source.
Ls 139-140. “there is no strong evidence, in general, for the necessity of micronutrient supplementation after injury”. Is this paper suggesting that it is important? Why the review if there is no necessity?
Ls 158-159. “During the rehabilitation phase, antioxidants, vitamins, minerals and other supplements are needed for the appropriate nutritional sup port.” Please be specific. Revise. What is meant by other supplements for example.
L195. Please provide supporting evidence that creatine intake is required when you are healthy.
L213. Please provide original sources that minerals have a beneficial effect on the performance of athletes.
L235. Are omega-3 fatty acids micronutrients? Please provide a supporting reference.
L290. “flavonoids and polyphenols”. Flavonoids are polyphenols. Please revise.
Ls 300-316. Please provide original sources. It seems just repeating information from Close et al (Ref 7).
Author Response
R: The title suggests that the role of micronutrients will be covered in the manuscript but there is also communication about leucine (an amino acid) and creatine, for example. Please reconsider the title.
A: The title of the manuscript has changed
R: The manuscript would benefit from a definition/description of micronutrients to be clear for the reader what nutritional elements are considered micronutrients. Is there consensus that creatine and leucine are considered micronutrients?
A: We rephrase the title of the manuscript
R: I feel that there is a lack of focus in the review and most of the information seems to be deal with effects of nutrients intake.
A: We rephrase the title and the aim of the manuscript
R: There seems to be a tendency not to use primary sources, e.g. ref 6 (L51). Please use original sources or clarify that review articles were referenced.
A: We corrected the reference.
R: Is recovery the same as rehabilitation. Recovery post-exercise is short lasting. Please be consistent in the manuscript.
A: The term recovery when used in place of rehabilitation, has been replaced
R: L23. “rich in nutrients”. Please clarify what kind of nutrients. Do you mean rich in particular micronutrients?
A: We corrected the sentence and clarified the meaning.
R: L25. “mechanisms of nutrient”. Do you mean micronutrients?
A: We corrected the sentence
R: L39. Reference 7 has no information on stress fractures and tendinopathy. Please check.
A: We include the appropriate reference.
R: L42. Ref 9. I suggest to replace with an English reference.
A: We included an English reference corresponding to sport sarcopenia
R: L44. Is there consensus that changes in muscle by prolonged injury are considered a sarcopenic event. Sarcopenia seems to be reserved mostly for aging-induced changes in muscle mass and function. Please reconsider.
A: We included a reference corresponding to sport sarcopenia.
R: L53. Ref 11 is on macronutrients but also cover leucine for example. Is there replication?
A: We removed the paragraph referring to leucine
R: L59. Considering the aim of the review, why is there no advice on required micronutrient intake.
A: We rephrase the aims of the manuscript. As there are no specific DRIs for athletes specific for the nutritional elements we are dealing with, beyond the current guidelines for the general public and the specific groups, no other intake guidelines are referred.
R: L64. No information in vitamins is presented in this section.
A: We include a sentence mentioning the anti-inflammatory effect of vitamins.
R: L69. What are “high-quality products”. Please clarify and provide an example of such a product.
A: We rephrase the sentence.
R: L73. “overproduction” to get oxidative stress but we normally always experience some level of oxidative stress. In addition, ref 15 does not provide experimental observations on overproduction of ROS. Please clarify/revise.
A: We remove the word overproduction. As Fiedor et al. write, that caretonoids help keeping the balance between antioxidant-reactive oxygen species.
R: Ls 77-89 is on aging and weight loss. The authors should focus their writing on rehabilitation and micronutrients.
A: We delete this sentence.
R: L92. Recently?, is 2013 recently?
A: We rephrase the sentence.
R: L95. “Flavonoids should be included in daily diet”. What is the recommendation considering it is suggest to have higher intake with severe injury. Please provide original references to support this assertion.
A: We rephrase the sentence. We also change the reference.
R: L106. Ref 20 is not an original source to support the statement. Please provide an original source.
A: We provide an original source for our statement.
R: Ls 139-140. “there is no strong evidence, in general, for the necessity of micronutrient supplementation after injury”. Is this paper suggesting that it is important? Why the review if there is no necessity?
A: We rephrase the sentence.
R: Ls 158-159. “During the rehabilitation phase, antioxidants, vitamins, minerals and other supplements are needed for the appropriate nutritional sup port.” Please be specific. Revise. What is meant by other supplements for example.
A: We rephrase the sentence.
R: L195. Please provide supporting evidence that creatine intake is required when you are healthy.
A: We rephrase the sentence.
R: L213. Please provide original sources that minerals have a beneficial effect on the performance of athletes.
A: We included the appropriate sources.
R: L235. Are omega-3 fatty acids micronutrients? Please provide a supporting reference.
A: We rephrase the head of the section, from “supplements of micronutrients” to “supplements of nutritional elements”.
R: L290. “flavonoids and polyphenols”. Flavonoids are polyphenols. Please revise.
A: We revise the sentence.
R: Ls 300-316. Please provide original sources. It seems just repeating information from Close et al (Ref 7).
A: We rephrase the sentences and include new references.
Reviewer 2 Report
The article entitled “The key role of micronutrients on sports rehabilitation” explores a very pertinent subject as injury is very common and safe and fast rehabilitation from it is very important for an athlete’s career. The authors organized the articles into chapters of micronutrients with functional affinities which should help the reader understand the role of these micronutrients in the body and their contribution to the recovery process.
Never the less, in each chapter, it is difficult to follow the Authors’ rationale. Maybe because most of the paragraph have English issues.
However, besides English issues, some of the sentences seem shortcuts where something is missing to follow the relation between cause and effect or intervention and outcome. This leads not only to a poor understanding of the effect of the supplements but also to important chemical inaccuracies.
In addition, or as a consequence, most of the supplements effects are poorly explored. For example: the review doesn’t explore the antioxidant properties of the antioxidant vitamins C and E, their different locations of action (cytosol/membranes) that results in the synergy between them and allows for cell stabilization, nor the synergy between vitamin E and selenium; It refers to the selenoenzyme glutathione peroxidase but does not refer catalase (iron) or superoxide dismutase (CuZn in cytoplasm or Mn in mitochondria); The role of leucine in protein synthesis is referred but not explored.
Although I believe this article should not be accepted unless the authors change the chemical language and that it needs major changes, here are some specific points:
Line 31- “Some injuries have no impact, but (…)”
Do the Authors mean: Some injuries have no impact on the future carrier or future quality of life of the athletes?
I think we can establish that all injuries have impact, even if only for a short period of time.
Lines 32-33 more information is necessary regarding the international championships considered (modalities/number/particular area of the globe if applicable) and during what period of time.
Line 42 bibliography gap – research gap?
Lines 53-54 has already extensively discussed – has already been extensively discussed
Lines 82-86 the intervention described is not clear
How much is the energy restriction?
The protocol followed macronutrients distribution of 30 % protein, 30 % carbohydrates, and 40 % fat.
Regarding to carotenoids: “enriched with high/low carotenoid vegetables/fruit and 40-60mL extra-virgin olive oil/polyunsaturated fatty acid based oil” – what was the approximate amount of carotenoids in the diets of the intervention groups? this sentence is not able to explain the study design or the cited study.
Line 91 – “benefits against exhaustive exercise” what kind of benefits during exhaustive exercise? Performance improvements? Perceived exertion? Other markers? Health markers?
Line 110 – consider changing to: In vitro studies have shown that vitamin D receptors (VDR) are expressed …
Line 112 (…) treatments have a positive impact on (…) – what kind of treatments? on muscle fibers in cells in vitro, (…) what do you mean?
Lines 116-118 – in terms of design what are the characteristics of the studies included in these meta-analysis? The following sentence refers to randomized controlled trials.
Lines 114-122 – were the studies controlled for exercise or included exercise intervention?
Line 125 – consider changing “(…) they tend to lead to oxidative stress, when in deficiency (…)” to: their deficiency tends to lead to oxidative stress
Lines 126-127 – “After injury (…) muscle atrophy.” The link between vit C, cytokines and muscle atrophy is not comprehensible.
Lines 129-130 – the reference to the role of Vit C in the hydroxylation of proline, and consequently collagen stabilization is not comprehensible.
Line 130 – “supplementation intake” either supplement intake or just supplementation
Line 131 - baseline values were obtained before or after the surgery?
Line 137 – what is “anti-cytokine effect”? There are lots of different cytokines. They respond differently to stimuli and induce different responses.
Line 150 – “injuries caused by bones”? “(….), intestines”? Can you be clearer?
Lines 152 – “(...) the inhibitory effect of cortisone on healing (…) can promote the cortisone-related healing in (…)” the sentence contradicts itself. Maybe the authors mean: “(...) the inhibitory effect of cortisone on healing (…) can revert the cortisone-related healing inhibition in (…)
Lines 172-173 - “(…) creatine supplementation (...) protective effect on muscle oxidative damage in cases of muscle inactivity.” relation between creatine and antioxidant properties is not clear.
Lines 181-183 - what is the consequence, in terms of muscle function, of having more GLUT4?
Chapter 3.2. Gelatin and vitamin C/ collagen is a subchapter of 3. Supplements of micronutrients. Gelatin is not a micronutrient and the effect of vitamin C on collagen synthesis has already been referred although superficially.
Line 208-209 – “Minerals, such as manganese (Mn), copper (Cu), zinc (Zn), iron (Fe) and selenium (Se) act as antioxidants, scavengers or proton donors, preventing ROS and repairing oxidative damage.” How can the minerals Mn, Cu, Zn, Fe or Se act as proton donors? This chemistry is not possible.
preventing ROS form doing what?
Alone these minerals are not scavengers and some of them, like coper and especially iron, are promoters of ROS generation – Iron promotes Fenton reaction.
I believe these are English issues but this sentence must be completely rephrased.
Line 211-212 – involving à involved
Line 229-232 – “More specifically, glutathione (…) selenoproteins.” – this sentence is not comprehensible.
Line 236 – “Unsaturated fatty acids as antioxidants, have a beneficial on lipid oxidation and immune-modulatory properties.” - Unsaturated fatty acids are not antioxidants. In fact unsaturated fatty acids are more susceptible to oxidation then saturated fatty, thus the probability of initiation and propagation of ROS uncontrolled reactions is higher. W-3 FA act as an unti-inflammatory agents and thus they end up reducing ROS generation.
Chapter 3.5. Leucine – the most important impact of leucine is its ability to trigger protein synthesis – this is not explored properly
Line 273 – diclofenac sodium – sodium diclofenac
Author Response
The article entitled “The key role of micronutrients on sports rehabilitation” explores a very pertinent subject as injury is very common and safe and fast rehabilitation from it is very important for an athlete’s career. The authors organized the articles into chapters of micronutrients with functional affinities which should help the reader understand the role of these micronutrients in the body and their contribution to the recovery process.
Never the less, in each chapter, it is difficult to follow the Authors’ rationale. Maybe because most of the paragraph have English issues.
However, besides English issues, some of the sentences seem shortcuts where something is missing to follow the relation between cause and effect or intervention and outcome. This leads not only to a poor understanding of the effect of the supplements but also to important chemical inaccuracies.
In addition, or as a consequence, most of the supplements effects are poorly explored. For example: the review doesn’t explore the antioxidant properties of the antioxidant vitamins C and E, their different locations of action (cytosol/membranes) that results in the synergy between them and allows for cell stabilization, nor the synergy between vitamin E and selenium; It refers to the selenoenzyme glutathione peroxidase but does not refer catalase (iron) or superoxide dismutase (CuZn in cytoplasm or Mn in mitochondria); The role of leucine in protein synthesis is referred but not explored.
Although I believe this article should not be accepted unless the authors change the chemical language and that it needs major changes, here are some specific points:
R: Line 31- “Some injuries have no impact, but (…)”
Do the Authors mean: Some injuries have no impact on the future carrier or future quality of life of the athletes?
I think we can establish that all injuries have impact, even if only for a short period of time.
A: We rephrase the sentence.
R: Lines 32-33 more information is necessary regarding the international championships considered (modalities/number/particular area of the globe if applicable) and during what period of time.
A: Further information were added
R: Line 42 bibliography gap – research gap?
A: We rephrased the sentence.
R: Lines 53-54 has already extensively discussed – has already been extensively discussed
A: We corrected the sentence.
R: Lines 82-86 the intervention described is not clear
How much is the energy restriction?
The protocol followed macronutrients distribution of 30 % protein, 30 % carbohydrates, and 40 % fat.
Regarding to carotenoids: “enriched with high/low carotenoid vegetables/fruit and 40-60mL extra-virgin olive oil/polyunsaturated fatty acid based oil” – what was the approximate amount of carotenoids in the diets of the intervention groups? this sentence is not able to explain the study design or the cited study.
A: We rephrase and corrected the sentences.
R: Line 91 – “benefits against exhaustive exercise” what kind of benefits during exhaustive exercise? Performance improvements? Perceived exertion? Other markers? Health markers?
A: We rephrase and corrected the sentences.
R: Line 110 – consider changing to: In vitro studies have shown that vitamin D receptors (VDR) are expressed …
A: We corrected the sentence.
R: Line 112 (…) treatments have a positive impact on (…) – what kind of treatments? on muscle fibers in cells in vitro, (…) what do you mean?
A: We corrected the sentences.
R: Lines 116-118 – in terms of design what are the characteristics of the studies included in these meta-analysis? The following sentence refers to randomized controlled trials.
A: We corrected the sentences.
R: Lines 114-122 – were the studies controlled for exercise or included exercise intervention?
A: We delete this study.
R: Line 125 – consider changing “(…) they tend to lead to oxidative stress, when in deficiency (…)” to: their deficiency tends to lead to oxidative stress
A: We corrected the sentence.
R: Lines 126-127 – “After injury (…) muscle atrophy.” The link between vit C, cytokines and muscle atrophy is not comprehensible.
A: The statement was omitted
R: Lines 129-130 – the reference to the role of Vit C in the hydroxylation of proline, and consequently collagen stabilization is not comprehensible.
A: This statement was omitted
R: Line 130 – “supplementation intake” either supplement intake or just supplementation
A: We corrected the sentence.
R: Line 131 - baseline values were obtained before or after the surgery?
A: They were obtained before surgery. We corrected the sentence.
R: Line 137 – what is “anti-cytokine effect”? There are lots of different cytokines. They respond differently to stimuli and induce different responses.
A: We corrected the sentence.
R: Line 150 – “injuries caused by bones”? “(….), intestines”? Can you be clearer?
A: We corrected the sentence.
R: Lines 152 – “(...) the inhibitory effect of cortisone on healing (…) can promote the cortisone-related healing in (…)” the sentence contradicts itself. Maybe the authors mean: “(...) the inhibitory effect of cortisone on healing (…) can revert the cortisone-related healing
inhibition in (…)
A: We corrected the sentence.
R: Lines 172-173 - “(…) creatine supplementation (...) protective effect on muscle oxidative damage in cases of muscle inactivity.” relation between creatine and antioxidant properties is not clear.
A: This sentence has been rephrased in order to better reflect the meaning of the original argument stated in the reference.
R: Lines 181-183 - what is the consequence, in terms of muscle function, of having more GLUT4?
Α: We include the beneficial effect of having more GLUT4 “increasing insulin sensitivity, exerting a beneficial effect on glucose homeostasis throughout the body and thus, increases glucose uptake into muscle”
R: Chapter 3.2. Gelatin and vitamin C/ collagen is a subchapter of 3. Supplements of micronutrients. Gelatin is not a micronutrient and the effect of vitamin C on collagen synthesis has already been referred although superficially.
A: We rename the chapter 3. As the reviewer correctly noticed the vitamin C on collagen synthesis has already been referred superficially above, so know it is explained in more detail.
R: Line 208-209 – “Minerals, such as manganese (Mn), copper (Cu), zinc (Zn), iron (Fe) and selenium (Se) act as antioxidants, scavengers or proton donors, preventing ROS and repairing oxidative damage.” How can the minerals Mn, Cu, Zn, Fe or Se act as proton donors? This chemistry is not possible.
A: We corrected the sentence.
R: preventing ROS form doing what?
Alone these minerals are not scavengers and some of them, like coper and especially iron, are promoters of ROS generation – Iron promotes Fenton reaction.
I believe these are English issues but this sentence must be completely rephrased.
A: We corrected the sentence.
R: Line 211-212 – involving à involved
A: We corrected the sentence.
R: Line 229-232 – “More specifically, glutathione (…) selenoproteins.” – this sentence is not comprehensible.
A: We corrected the sentence.
R: Line 236 – “Unsaturated fatty acids as antioxidants, have a beneficial on lipid oxidation and immune-modulatory properties.” - Unsaturated fatty acids are not antioxidants. In fact unsaturated fatty acids are more susceptible to oxidation then saturated fatty, thus the probability of initiation and propagation of ROS uncontrolled reactions is higher. W-3 FA act as an unti-inflammatory agents and thus they end up reducing ROS generation.
A: We corrected the sentence.
R: Chapter 3.5. Leucine – the most important impact of leucine is its ability to trigger protein synthesis – this is not explored properly
A: We removed the paragraph referring to leucine, as part of another reviewer’s comments.
R: Line 273 – diclofenacsodium – sodiumdiclofenac
A: We corrected the word.
Round 2
Reviewer 1 Report
L64. I suggest to change "to recover glycogen muscle damage" to "to recover glycogen content and from muscle damage".
L130. I suggest to change "it benefit" to "it benefitted".
l132. I suggest to change "In details a meta-analysis" to "A meta-analyis".
L226. I suggest to change "as are" to "as they are".
L228. I suggest to change "pick athletic" to "peak athletic".
L268. I suggest to change "sodiumdiclofenac" to "sodium diclofenac".
L308. Please provide the Close reference.
Author Response
R: L64. I suggest to change "to recover glycogen muscle damage" to "to recover glycogen content and from muscle damage".
A: We corrected the sentence
R: L130. I suggest to change "it benefit" to "it benefitted".
A: We corrected the sentence
R: l132. I suggest to change "In details a meta-analysis" to "A meta-analyis".
A: We corrected the sentence
R: L226. I suggest to change "as are" to "as they are".
A: We corrected the sentence
R: L228. I suggest to change "pick athletic" to "peak athletic".
A: We corrected the sentence
R: L268. I suggest to change "sodiumdiclofenac" to "sodium diclofenac".
A: We corrected the sentence
R: L308. Please provide the Close reference
A: We correct the reference
Reviewer 2 Report
The authors addressed most of the suggestions. I think the chemical inaccuracies were eliminated. The option was to eliminate or simplify the sentences that were exploring the subject from a more molecular point of view. Overall, the article is consistent.
Some minor suggestions:
Lines 37… are the authors restricting information to track and field championships
Line 152 – “supplementation intake” either supplement intake or just supplementation
Line 153 – what is “cytokine production” Which cytokines were studied by the author cited? There are lots of different cytokines. They respond differently to stimuli and induce different responses.
Chapter 3.3 sentences repeat the same idea over again.
Line 239 – “bone health”? the paragraph is general why focus on bone?
Please check the text for typos.
Author Response
The authors addressed most of the suggestions. I think the chemical inaccuracies were eliminated. The option was to eliminate or simplify the sentences that were exploring the subject from a more molecular point of view. Overall, the article is consistent.
Some minor suggestions:
R: Lines 37… are the authors restricting information to track and field championships
A: Further information were added
R: Line 152 – “supplementation intake” either supplement intake or just supplementation
A: We corrected the sentence
R: Line 153 – what is “cytokine production” Which cytokines were studied by the author cited? There are lots of different cytokines. They respond differently to stimuli and induce different responses.
A: The exact cytokines that were studied are now clearly stated.
R: Chapter 3.3 sentences repeat the same idea over again.
A: We corrected and rephrase the paragraph.
R:Line 239 – “bone health”? the paragraph is general why focus on bone?
A: We rephrased the sentence
Please check the text for typos.